# An Analysis of the Demand-Side, Platform-Based Collaborative Economy: Creation of a Clear Classification Taxonomy

Tünde Zita Kovács *, Forest David, Adrián Nagy, István Szűcs and András Nábrádi

Institute of Applied Economic Sciences, Faculty of Economics and Business, University of Debrecen, 4032 Debrecen, Hungary; forestdavid5@gmail.com (F.D.); nagy.adrian@econ.unideb.hu (A.N.); szucs.istvan@econ.unideb.hu (I.S.); nabradi.andras@econ.unideb.hu (A.N.)
* Correspondence: kovacs.tunde.zita@econ.unideb.hu; Tel.: +40-727-599-263

**Abstract:** The rapid proliferation of the demand-side, platform-based collaborative economy and its various forms have been an aspect of everyday life for over a decade. However, despite the platform's popularity, the descriptions and names attributed to the demand-side, platform-based collaborative economy are often used interchangeably and can be ambiguous, resulting in confusion among researchers and practitioners. This study examines the articles published during the previous ten years, which have aimed not only to define, but also to classify, the collaborative economy according to various criteria. After reviewing the existing classification criteria, this article presents a clearer taxonomy of the multiple forms of the collaborative economy by distinguishing service providers' groups on their respective attributes. Our review and analysis have both theoretical and practical importance. Regarding the latter, our research will help managers and government officials alike in rethinking the taxation and subsidizing policies related to the various demand-side, platform-based collaborative economies and in preparing national and international consultations and conventions. This study aims to define the demand-side, platform-based collaborative economy and integrate the concept into various economic activities, providing a new and valuable contribution to the literature.

**Keywords:** platform-based economy; sharing economy; collaborative economy; collaborative consumption; demand-side

## 1. Introduction

During the last ten years, the number of Internet platform-based enterprises has dramatically increased, creating online structures that have facilitated a wide range of human activities [1]. The appearance and the development of platform-based businesses have been influenced by several factors, including social, informational, technological and economic factors [2–5]. The development of the Internet, widespread appearance of mobile applications, growth in big-data analysis, appearance of artificial intelligence-based applications, rapidly changing consumer needs and habits, increasing economic inequalities and economic crises have all contributed to the formation and advancement of platform businesses [6]. The platform-based business model described in this paper is presented as a demand-side economic activity, built on trust between the service recipient and the provider through an information technology organization serving as an intermediary in the digital marketplace [7]. This new business model evolved through the influence of the previously mentioned factors and is commonly interchangeably referred to as the collaborative economy [5,8], the collaborative consumption [9–11] or the sharing economy [12]. The sharing economy is probably the most common of the three terms used by researchers and users of the services. However, there are clear distinctions between the three terms commonly used to describe this demand-side, platform-based business model.

"The sharing economy has become a catch-all label with strong normative underpinnings" [13] (p. 1). Serving as a general term, the sharing economy (SE) does not denote the

same concept as its less popular alternative names. While the SE is an excellent and well-known construct used in the literature [14–16], we proposed the collaborative economy (CoE) as the more broad umbrella construct encompassing both the SE and collaborative consumption (CC). Therefore, the SE and CC are two similar but not identical economic and social systems that refer to the Internet platform–based sharing (on a temporary or permanent basis) of human and physical resources. The fundamental difference between the two social systems is the service provider's legal status: private individual (peer-P) or company (business-B). SE systems can be best explained as private individuals becoming service providers themselves through using an online Internet platform as an intermediary, reducing the conventional pipeline (value chain–based) businesses' competitive advantages [17].

In contrast, CC systems are larger businesses with more formal financial systems and financial reporting requirements than SE systems. However, like SE operators, CC operators also use an established online intermediary as the linchpin connecting customers to their service, reducing the traditional advantages of having a well-established value chain. Thus the CoE has revolutionized entire industries, as not only have former consumers (now SE providers) turned into service providers themselves, but many former businesses (CC providers) have also transformed their business model by implementing the use of the CoE, thus generating new sophisticated competitors to more acutely rival traditional service providers. The CoE concept, described by Schumpeter [18] as creative destruction, has now created new global business juggernauts, among which the most notable include: Uber (ride-sharing), Airbnb (flat sharing), Task Rabbit (freelance lab), Kickstarter (crowd-funding), Waze (social online mapping/traffic), Etsy (an online marketplace for handmade products), ThredUp (second–hand clothes), and WeWork (co-working offices) all serving as intermediaries, whereby CC and SE actors risk the majority of the capital with the broader CoE firm providing the platform.

The purpose of this paper was to develop a new taxonomy classification to explain the CoE and improve on the lack of current consensus regarding the terms and spheres of the CoE's activity. To the extent possible, our descriptions will be confirmed by examples, facilitating the understanding of SE and CC concepts, and identifying their similarities and distinguishing their differences. Subsequently, we will determine and classify the types of demand-side, Internet platform-based companies as opposed to value chain–based companies, which operate on traditional principles and are mostly supply-oriented [19].

## 2. Literature Review

The question of economic sustainability has also ushered in the winds of change [20,21]. Besides, the global network, which has become easily accessible to anyone, has brought about the birth of local and international communities; meanwhile, artificial intelligence (AI) and big-data analysis have created more generous supply and demand harmonization. It may seem attractive for the business that the CoE is nourished by new cultural and social ideologies and a new respect for environmental protection considerations. The findings of Wallenstein—Shelat [22] on consumption have revealed this economic model is driven more by the prevailing economic motivation; e.g., relatively low price and better service than conventional providers, rather than changes in human demands and needs. The vast majority of actors engaged in the CoE battlefield would be classified as SE actors and are microenterprises or simple taxpaying entrepreneurs, making it historically challenging to discover the CoE's true magnitude [23]. According to one PwC study [24], CoE companies generated sales revenue of $15 billion in 2013 and are estimated to increase to $335 billion by 2025. For example, Uber Technologies generated over $14 billion alone in 2019 and over $11 billion in the pandemic 2020 year. However, as many of these firms become publicly traded companies such as Uber Technologies and Airbnb, they are required by law to report extensive financial documentation just as Google, Apple, BMW, or any other publicly-traded company would. Besides, private firms engaging in the CoE market, such as Vacation Rental by Owner (VRBO), are increasingly forced by local governments to

report all transactions for local tax collection purposes, thus providing detailed information used to determine the total impact of CoE. However, assessing SE or CC's impact may remain difficult as such segment data is generally not required to be published.

In the international literature, various alternative terms are frequently used for the CoE (Table 1), which impedes understanding the phenomenon. Such terms often overlap each other or are interchanged at random [16]; adjacent notions are recommended.

**Table 1.** The various terms of the collaborative economy (CoE).

| Term | Literature |
| --- | --- |
| sharing economy | Benkler [25] |
| | Heinrichs [26] |
| | Botsman—Rogers [9] |
| collaborative consumption | Hamari et al. [10] |
| | Schor—Fitzmaurice [11] |
| peer-to-peer based sharing | Schor [27] |
| crowd-based capitalism | Sundararajan [28] |
| gig-economy | Friedman [29] |
| | Gyulavári [30] |
| connected consumption | Schor—Fitzmaurice [11] |
| access economy | Eckhardt—Bardhi [31] |
| on-demand economy | Cockayne [32] |
| | Selloni [33] |
| gift-economy | Eisenstein [34] |
| circular economy | Ness [35] |

Each of the above-listed terms describes the same broad model but underlining different perceived attributes. These terms point to specific vital aspects of this economic tendency, which are the following: idle capacity, network, community, trust, sharing, swapping, lending and demand.

The SE and CC involve the collective creation, production, trade, traffic and consumption of goods and services on the part of organizations and individuals. The SE and CC constitute diverse market structures, in which trading between economic actors is realized in various market configurations, by "creating an opportunity to make use of the hitherto compiled capacities which mostly lie idle and unused" [36] (p. 109), with the help of sharing and reuse.

This paper aims to present the CoE through its names and definitions. Our purpose was to underline the lack of consensus regarding the terms and spheres of the CoE's activity. To the extent possible, our descriptions will be confirmed by examples, facilitating the understanding of SE and CC concepts, and identifying their similarities and distinguishing their differences. Subsequently, we will determine and classify the types of CoE in contrast to value chain-based companies, which operate on traditional principles and are mostly supply-oriented [19].

## 3. Materials and Methods

When conducting this study, we first analyzed the international literature and then systematized the SE and CC online business structures. Our systematization is based on the classification presented in Botsman—Rogers [37], which we later extended according to other authors' perspectives (Appendix A). After analyzing the differences and distinguishing features between the SE and CC, we created a new classification taxonomy based on a clear grouping without overlaps.

In the first part of the study, we highlighted differences between value chain-based and platform-based businesses, using existing literature on the topic. Subsequently, we assessed the literature, knowing it was impossible to evaluate every study and examined the studies that defined the CoE and aimed to establish criteria for its classification and grouping. Based on Paul et al. [38] TCM framework, wherein T refers to theories, C refers

to contexts, and M refers to methods, the methodology used in our paper is summarized in (Table 2). Up-to-date literature sources were collected using mainly Google Scholar, Web of Science, SpringerLink and European Commission (EC) databases. Technical literature databases were searched with the search terms listed in (Table 2). In searching for literature sources, the results were mainly focused on the years between 2010 and 2020. As a next step, a content analysis was performed on the selected literature. We also reviewed the bibliography of the literature collected in this way, and we expanded the range of relevant literature using the snowball method. The resulting database was filtered around the following classifications criteria: type of performed activities, providers' person and market structure, mode of exchange and business model. For this reason, we are focusing only on the articles which met the selection criteria. After applying the exclusion criteria, 59 articles were included in the review (Appendix A).

**Table 2.** Summary of the methodology used in the study.

| Research Objective |
|---|
| Developing a new and improved method of grouping and classification of CoE in four criteria: |
| 1. Classification of the type of performed activities (core activities)<br>2. Market structure (initial provider)<br>3. Mode of exchange<br>4. Business model |
| **Initial Inclusion Criteria** |
| The scientific databases Google Scholar, Web of Science, SpringerLink and European Commission (EC) were searched for relevant scientific publications. |
| **Setting the Inclusion Criteria** |
| 1. Search terms: collaborative economy; sharing economy; collaborative consumption; circular economy; access economy; peer-to-peer based sharing; crowd-based capitalism; the gig economy; connected consumption; platform economy; multi-sided platforms; on-demand economy; gift-economy; peer economy.<br>2. Time frame: 2010–2020<br>3. Transaction both with and without transfer of property ownership;<br>4. Providers person: legal entities and private individuals<br>5. Business structure: demand-side platform-based businesses |
| **Applying the Exclusion Criteria** |
| After the reading of titles and abstracts, articles were removed that were NOT focused on the following criteria: |
| 1. Type of performed activities in the CoE;<br>2. Market structure (initial provider) in the CoE;<br>3. Transactions mode of exchange in the CoE;<br>4. Classification of demand-side platform-based business |
| **Content Analysis** |
| In-depth analysis and classification of selected papers according to the four criteria of research objectives.Compared findings of selected papers on criteria of research objectives. |
| **Discussion and Conclusion** |

The new classification criteria are made up of the following steps: (1) we disregarded the various names of the CoE, which are often adjacent and overlapping, and based our classification on the type of activity performed; for example, transport, accommodation, etc. Each type of activity may have several implementation methods, but it must be linked to the same core activity; e.g., car-sharing, ride-sharing, car renting are related to transportation. (2) We distinguished two major groups, based on who is responsible for providing the goods or the service: a legal person, i.e., a business, or a private individual, i.e., a peer, who possesses limited means. At that point, we defined the concepts of CC and SE. (3) Another question was whether the change in ownership occurs after a transaction is concluded, or not thus exemplifying the need for having two categories, distinguishing

the redistribution system from the product-service system. (4) In the final stage of the systematization, we included each of the categories and groups under the umbrella of demand-side, platform-based CoE.

The systematization and classification of the fundamental concepts of CoE are demonstrated in the following logical scheme:

1.    The separation of the pipeline business model from platform-based business models.
2.    The definition of the demand-side, platform-based CoE.
3.    The categorization of the demand-side, platform-based CoE

The redistribution market is characterized by the change in ownership from supplier to buyer. In contrast, the product-service system is characterized by the temporary use of a product. Both the redistribution market and product-service systems can be further divided into two subgroups based on who is responsible for providing the service—a legal business entity (B) or a private individual (peer-P):

- Supplier is a legal business entity where transactions are characterized as Business to Peer (B2P) or Business to Business (B2B).
- Supplier is a private individual or peer where transactions are characterized as Peer to Peer (P2P) or Peer to Business (P2B).

Business enterprises are legal entities that carry out economic activities with their own or external assets and labor, for financial gain, in the long run, and with risks [39]. Companies are required to prepare a statement about the financial position and to pay taxes. In contrast, a private individual is a natural person who has only limited means and is not obliged to prepare an account with a balance sheet. When legal business entities are suppliers in the transaction (B2P, B2B), we consider this part of the CC. The SE involves those transactions where a private individual is a supplier (P2P, P2B). This classification is summarized in (Figure 1):

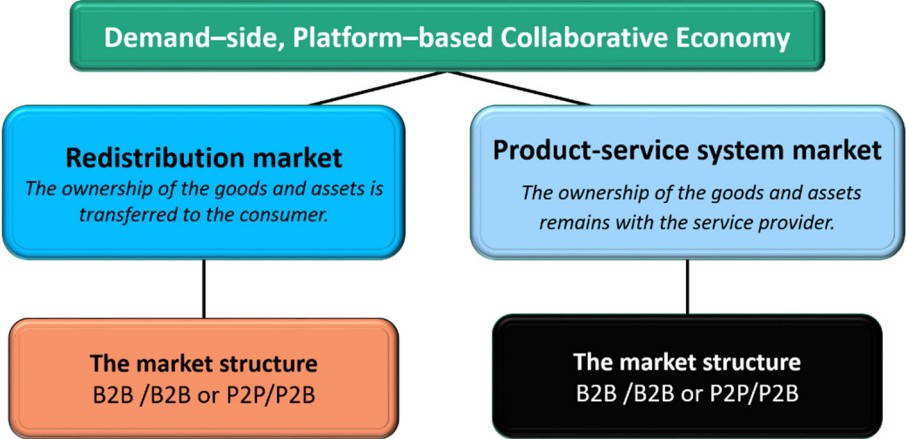

**Figure 1.** The fundamental separation of the main categories of demand-side, platform-based CoE.

We demonstrated both the product/services branches' characteristics, based on the categories: redistribution market and product-service system market and its division into groups, and considered their respective manifestations through practical examples.

## 4. Results

### 4.1. The Separation of the Pipeline Business Model from Platform-Based Business Models

For many decades, companies have operated according to a well–defined scheme. Such companies are called traditional, value chain–based companies, where value creation is part of a linear series of activities, and the value itself moves from the left side, i.e., the costs of the company, to the right side, i.e., the revenues of the company [40]. A company's value chain is a series of activities by which a greater value is created from the resources through various transformations. This greater value contributes to both meet the demands

of consumers and yielding a profit for the businesses [41]. This reveals that the company's internal activities predominately control the process of value creation. The economy of the value chain–based companies depends on resources, and its strength lies in possession of (a) rare or even exclusive resource(s) or capacity(s) which cannot be copied [42]. Its primary aim is to optimize the value chain processes, from supply purchase through production to sales and service. The company's value to individual consumers is reached by maximizing the efficiency of the company's inner activities while ensuring protection against the factors affecting competitiveness that Porter [43] termed the five forces. In a supply–oriented economy, companies can dominate the market by controlling the resources, increasing efficiency fiercely, and reducing the five forces' challenges [44].

In contrast, in the platform-based business model, the primary resource is producers and consumers' network. What the company owns is matters less than the resource of the cultivated network. In the process of value creation, the platform connects distinct types of affiliated actors, operating as an intermediary, enabling the interaction between two or more independent groups, usually producers and consumers [45,46]. The process of value creation has two phases: building connections and "manufacturing" transactions. The platform aims to maximize the expanding ecosystem's value, which is circular, recurring and feedback–oriented. To make this possible, "a sort of network effect is manifested concerning the actors of the single segments" of the platform, "which leads to the consequence that consumers benefit from the expansion of the other segment. In the platforms, one segment's demand depends on the size of the other segment" [47] (p. 2). In the case of platform-based companies, there are costs and revenues in both sides' segments, and to achieve the desired network effect, one side (segment) is often subsidized [48]. In demand-side economies, the five competitive forces described by Porter can act as accretive external forces when the power of the consumers or the suppliers is used as a resource [44].

Developing a mobile application, creating a website, or building a web store is not the same as the platform as a business model. The platform is not just a piece of software or technology. The company still operates on a value chain basis; the value creation process does not change; the created value remains in the forms of goods or services. They create only an alternative interface through which they can address their partners. The platform-based business model is the holistic approach of platform business that focuses on building connections, facilitating the interaction between two or more participant groups. The platform's roles are matchmaking, to aid participants by matching them together, to provide the technology to facilitate transactions and establishing the governance structure, setting rules and standards to build trust, maintain quality, and achieve the desired network effect [49].

Our conclusion is that value chain–based companies operate on supply and value chain principles, while CoEs predominantly operate on demand and network–building and using Internet platforms. In the case of capacity–restricted goods, e.g., physical means, capital, labor force, there is an indirect network effect, which means that the value of a service increases for one user group when a new user or a different user group joins the network [50]. Consequently, the indirect network effect itself is a reasonable and efficient driving force for expanding a business. Demand-side and platform-based companies create value by promoting interaction between two or more customers or segments.

### 4.2. The Definition of the Demand-Side, Platform-Based Business Model
The Definition of the CoE

The international literature does not aim to create a consensus regarding names and definitions [17]. In Botsman—Rogers [9], the expression CC is preferred to SE. They believe that the excessive consumption of goods, which characterized the 20th century and was financed mostly by debt-spending, is replaced by collaboration–based consumption, motivated by reputation and prestige. Hyper-consumption is typically supported by advertising, which drums the advantages of possession and ownership into consumers'

heads. In contrast, CC is community-oriented, strengthening the sense of belonging through sharing and collaboration [9].

CC offers new opportunities for sustainability, supporting sustainable development and environmental protection, through the "holy trinity" of efficiency, consistency and sufficiency [26].

According to Schor [2], sharing activities are self–defined and confirmed by the media. Anybody can decide whether the action they perform belongs to the category of the sharing economy or not. There are four broad spheres of activities: recirculation of goods, increased utilization of durable means, exchange of services and sharing of productive means.

Sundararajan [28] and Martin et al. [51] highlighted the importance of sharing, especially in the promising and leading P2P markets, and urged users to continue their economic activities. Sundararajan [28] and Stephany [52] analyzed the shift in the consumption model from possession to sharing, which reduces the need for possessing the means.

"The expression SE is commonly used to indicate a wide range of digital commercial or non-profit platforms facilitating exchanges amongst a variety of players through a variety of interaction modalities (P2P, P2B, B2P, B2B) that all broadly enable consumption or productive activities leveraging capital means (money, real estate property, equipment, cars, etc.) goods, skills, or just time" [53] (p. 22).

The European Commission has defined the CoE as follows: "( . . . ) the term CoE refers to business models where activities are facilitated by collaborative platforms that create an open marketplace for the temporary usage of goods or services often provided by private individuals. The collaborative economy involves three categories of actors:

(i)     service providers who share means, resources, time and/or skills—these can be private individuals offering services on an occasional basis ('peers') or service providers acting in their professional capacity ('professional services providers');
(ii)    users of these; and
(iii)   intermediaries that connect—via an online platform—providers with users and that facilitate transactions between them ('collaborative platforms').

CoE transactions generally do not involve a change of ownership and can be carried out for-profit or not-for-profit." [8].

Szegedi [54] claims that sharing-based business activities have four primary characteristics:

1.    The activity is implemented through a website, an application or an online platform.
2.    It enables P2P transactions.
3.    It provides temporary access to goods and services without a change of ownership. This characteristic excludes the sale of second–hand goods and online market trading from the umbrella of the SE.
4.    Unused means, services, skills or resources are put to use.

To summarize, the various authors and sources of the topic have emphasized and highlighted different elements regarding companies belonging to the CoE's umbrella. Their definitions and descriptions of models contain similarities and differences, and overlapping also often occurs. Botsman—Rogers [9] emphasize collaboration-based consumption, Sundararajan [28] and Martin et al. [51] underline the new way of establishing connections, while Sundararajan [28] and Stephany [52] promote the sharing of valuable idle capital instead of possession, highlighting the importance of recognizing the barriers posed by superfluous control. Codagnone et al. [53] refer to it as a commercial and non–profit platform-based business, where sharing or trading can be implemented in various forms. Heinrichs [26] believes the benefits of this new economic model lie in the field of sustainability and environmental protection. Szegedi [54] insists that sales activities do not belong to the SE's umbrella, nor does trade, unless realized in a P2P form via a digital platform. According to the European Commission's interpretation [8], the transactions in the SE do not involve a change of ownership, and professional service providers are not excluded.

The above-mentioned definitions are taken from only a fraction of the international and Hungarian literature. We aimed to underline the lack of consensus regarding the CoE's activity's names and spheres. Most of the authors from the selected papers arguing that the CoE means sharing idle capacities only by providing temporary access to them; the sharing process excludes the possibility of a sale (Table 3).

**Table 3.** Transaction mode of exchange in the CoE.

| Selected Articles and Studies | Transfer of the Ownership (Sale and Purchase) | Without the Transfer of the Ownership (Short Term Access) |
|---|---|---|
| Schor [2] | - | √ |
| Wirtz et al. [6] | √ | √ |
| Botsman—Rogers [9,37] | √ | √ |
| Hamari et al. [10] | √ | √ |
| Schor—Fitzmaurice [11] | - | √ |
| Valant (EC) [8] | - | √ |
| Ranjbari et al. [7] | - | √ |
| Kumar et al. [17] | - | √ |
| PwC [24] | - | √ |
| Martin [16] | √ | √ |
| Sundararajan [28] | - | √ |
| Selloni [33] | - | √ |
| Acquier et al. [13] | √ | √ |
| Sutherland—Jarrahi [46] | - | √ |
| Heinrichs [26] | - | √ |
| Martin et al. [51] | √ | √ |
| Stephany [52] | - | √ |
| Codagnone et al. [53] | √ | √ |
| Szegedi [54] | - | √ |
| Haase—Pick [55] | - | √ |
| McDonald [56] | - | √ |
| Rauch—Schleicher [57] | - | √ |
| Richardson [58] | - | √ |
| Winterhalter et al. [59] | √ | √ |
| Edbring et al. [60] | √ | √ |
| Belk [61,62] | - | √ |
| Bardhi—Eckhardt [63] | - | √ |
| Matzler et al. [64] | √ | √ |
| Frenken—Schor [65] | √ | √ |
| Gerwe—Silva [66] | - | √ |
| Frenken [67] | √ | √ |

In contrast, others claim that it also includes reusing goods via a sale and purchase transaction. Similar to the opinion of Wirtz et al. [6], Botsman–Rogers [9,37], Hamari et al. [10], Martin [16], Acquier et al. [13], Codagnone et al. [53], Frenken [67], Martin et al. [51], Winterhalter et al. [59], Edbring et al. [60], Matzler et al. [64] and Frenken–Schor [65], we take a position that there is a change of ownership in redistribution market transactions. Maximum identification with the opinion Plewnia–Guenther "sharing products on the redistribution market means recirculation of good by selling, trading, bartering, swapping or giving them away for free." [68] (p. 574), which involve the transfer of ownership. It also complies with the requirements of sustainability.

Some researchers suggest that trade or sales can be implemented only between equal parties (peers), e.g., Schor [2], Belk [61], Gerwe–Silva [66], while others do not exclude other formats for the execution of transactions, e.g.,:Codagnone et al. [53], Laurenti et al. [69], Choi et al. [70], Olsen–Kemp [71]. The principle question on which there is more or less consensus is the presence of a digital network and applications and platforms. Otherwise, current research thought would not consider these businesses to be part of the Internet platform-based economy if these are not present.

However, what exactly does the term CoE mean? After reviewing the results of the international literature and eliminating the overlaps, we created a definition of the CoE:

> The CoE is a demand-side, platform-based economic activity generated by consumers, built on trust and operated by information technology organizations in digital markets. The consumers' needs are met on an online Internet platform by providing immediate access to new or underutilized goods, services or money, with or without the transfer of ownership. Information is provided by multiple third parties on digital marketplace product or service, whereas the marketplace operator enables direct interaction between the participant groups. (Authors' elaboration based on Kovács et al. [72]; Nábrádi–Kovács [73])

This section, divided by subheadings, provides a concise and precise description of the experimental results, their interpretation, and the practical conclusions are drawn.

### 4.3. The Main Categories of the CoE

To properly classify the platforms, we used the international literature sources, which rely on a wide–range of academic debates, and then systematized the SE and CC companies. Our systematization is based on the classification of Botsman–Rogers [37], which was upgraded and clarified by other authors. Platform-based companies, which include online stores, are familiar with most everyone. One can hardly find a shop that does not operate an online store where customers can browse, select and purchase a product without leaving their homes. When platforms enter the market of the value chain-based companies, they often develop a competitive advantage [44]. However, a store or a service provider will not be classified as having the SE or the CC business model only by having an online interface.

#### 4.3.1. The Classification of Demand-Side Platforms

To classify demand-side platforms, we compared several authors' viewpoints. We started with the classification of Botsman–Rogers [37], who distinguished three groups: the product-service systems market, redistribution markets and the collaborative lifestyles market.

- The product-service systems market is use–oriented, where the ownership of a product or goods remains with the service provider, who sells only the product's function through modified selling channels; examples include equipment finance (leasing), sharing, or pooling [74]. Consumers enjoy the benefits of the product without owning it. Car sharing, co-working offices and P2P equipment–rental are the best–known services of the product-service systems market.
- In the redistribution market, people exchange goods they do not want to use any longer. Due to many platform-based applications, this can happen not only locally but also globally. Of course, other products can be exchanged too, and selling is possible at prices that are often lower than traditionally provided by businesses operating outside of the SE.
- In the collaborative lifestyles market, the goods, which are shared, are usually not tangible and include time, money, know-how, skills, venues or space.

However, the Botsman–Rogers [37] grouping does not make a sharp distinction between these categories, since, as the authors also acknowledge, the collaborative lifestyles market, as its name says, denotes a lifestyle, which can include the product-service systems market, as well as redistribution markets, thus creating a hybrid market of the demand-side, Internet platform–based groups. Besides, they claim that the SE does not include activities in which the service provider/promoter is not a private individual but a business.

The demand–side Internet platforms have three main groups, according to Frenken [67] too: peer–to–peer, access-based and circular groups. Neither does Frenken make a sharp distinction: each group can partially cover another one. Therefore, there are also three subgroups: the second–hand economy, the demand-side economy and the service provider economy. Furthermore, there is the SE at the intersection of all three trends. The SE can best be realized between peers (P2P), according to Frenken, too; this occurs when they

grant each other temporary access to their idle capacity and thus make better use of an underutilized physical asset. However, he adds that the interaction between users can be realized in peer-to-peer form and B2P, B2B and P2B form too when a business is on the sender or the receiver side of a transaction. Internet platforms, usually in an application or a website, are essential for transactions [67].

In contrast, Wirtz et al. [6] make a far sharper distinction when grouping the activities performed on Internet platforms. They interpret the transaction according to two dimensions:

- Ownership, i.e., whether the access is only temporary; e.g., short–term rental, quick right of us, or the transaction is accompanied by the transfer of ownership, e.g., by selling, vending;
- The nature of the service provider; i.e., whether the provider of the means or the resources is a private individual or a legal entity.

4.3.2. Principles Distinguishing Attributes between the Redistribution Market and Product-Service Systems Market

To sum up, the viewpoints mentioned above, we determined only two categories of CoE (Figure 1):

1. Redistribution market
2. Product-service systems market

The redistribution market coincides with the category described in Botsman–Rogers [37], where it has a similar name, as well as with the circular economy of Frenken [67], where transactions are concluded with the transfer of ownership [6]. The product-service systems market involves the two other market types of Botsman–Rogers [37], i.e., the product-service system market and the collaborative lifestyles market. It also includes the other two types described in Frenken [67]: the peer-to-peer economy and the access–based economy, although in the case of these two market categories, ownership is not transferred [6], and consumers get only limited access to goods.

The differences between these two market categories are demonstrated in (Table 4). The most crucial difference between these two demand-side and platform-based market categories is the transfer of ownership or the lack of ownership transfer. The redistribution market focuses on the product at the customers' disposal in various forms, and the income originates from the sale of the product. After selling a product, its ownership is transferred from the service provider to the consumer. The service provider's task is to advertise the goods to be sold to attract consumers' attention. The product-service systems market focuses on the service, which means that service providers have mostly homogeneous spheres of activity, and the ownership is not transferred at the end of a transaction, since what customers purchase is a service that is limited in time or access to a product which is also limited. Service providers, both private individuals and companies, need to perform their activities at a high-quality level—a less essential requirement in redistribution platforms.

**Table 4.** Comparison of the redistribution market and the product-service systems market.

| Redistribution Market | Product-Service Systems Market |
|---|---|
| Product-oriented | Service-oriented |
| Ownership of the goods is transferred to the customer | Ownership of the goods remains with the service provider |
| Products are heterogeneous within the company | Products are relatively homogeneous within the service provider company |
| Face-to-face interaction is not required | Face-to-face interaction with consumers is essential |
| The quality of the service is secondary | The quality of the service is essential |
| Core marketing activities can be executed through suppliers | Core marketing activities cannot be executed through suppliers |
| The supplier has low risks associated with their involvement or assets due to the transfer of ownership | Service providers take high risks; they risk their goods due to the personal nature of the transaction |

Source: Authors' elaboration based on Kumar et al. [17].

Demand-side platforms are related to two economic activities: selling and renting. Selling goes with the transfer of ownership while renting the goods' ownership remains at the service provider or lender. To extend the findings of Wirtz et al. [6], we distinguish the groups of service providers in the case of CoE (Figure 2), the SE and CC. Furthermore, to separate these two from each other, we use a novel approach.

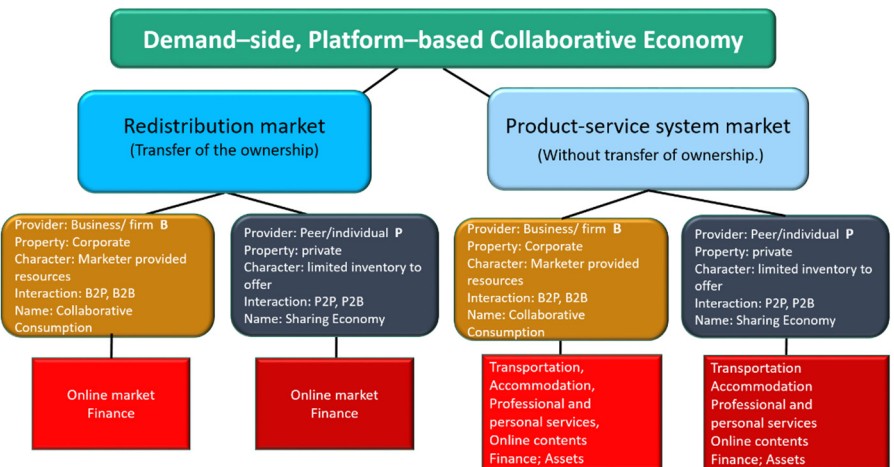

**Figure 2.** The classification of CoE; Source: Authors' edition based on Kovács–Nábrádi [75] and Nábrádi–Kovács [76].

## 5. Discussion and Practical Examples of Categories and Groups

### 5.1. The Segmentation of Redistribution Markets

The highlighted elements of the redistribution markets are presented in (Table 5), whereas the market structure, the basis of separation, and the type of performed activity are visible:

**Table 5.** The groups and characteristics of the redistribution market.

| Name | Mode of Exchange: Redistribution Market | |
|---|---|---|
| Market structure | Collaborative consumption, CC in the redistribution market | Sharing economy, SE in the redistribution market |
| Provider's person: Who is responsible for providing the goods or the service? | Legal entity (business)—a company with a balance sheet detailing its means and assets. The transactions of product-service are executed in B2P and B2B constructions. | Private individual (peer)—who has private property only. The transactions of product–service are executed in P2P and P2B constructions. |
| Type of performed activity | 1. Online Market<br>2. Finance | |

#### 5.1.1. CC in the Redistribution Market (B2B, B2P)

Our grouping is based on the fields in which service providers operate. We identified two types of activities in the CoE redistribution market: online markets and the financial sector (finance).

The online market is today's most popular vending channel, known and used all over the world. It can operate locally, at a national level or on a global scale. It is characterized by a broad product supply and consequently an enormous customer base, who tend to purchase as soon as they find their desired products. It includes the sale of new products and products no longer used by the supplier. Some of the most significant examples of online market places are Amazon, Alibaba, Taobao and Jumia.

The term "financial sector" refers to online crowdfunding platforms, where companies aimed at making investments offer funds in exchange for shares (equity financing), for ex-

ample, CircleUp, SeedInvest, Tőkeportál.hu. In contrast, other crowdfunding applications and websites focus on one sector, like agriculture: FarmFundr, Harvest Returns, the energy industry and green energy: SooofInc, Solar Share, Wien Energie. Details of the financial sector are presented in the section on the product-service system.

It is essential to highlight that the service provider or the business activity performer is always a business enterprise in CC. Therefore, it is a company that offers its products or means (funds) via CoE, and transactions are concluded with ownership transfer. In the case of finance, the company obtains shares of the financed economic activity.

### 5.1.2. SE in the Redistribution Market (P2B, P2P)

In the CoE redistribution market, where the service provider of economic activities is a private individual (peer), we also identified two fields as above: the online market and the financial market (finance). In this case, the crucial part of online markets develops locally, for example, Facebook Marketplace, Publi24 and Meska. At the same time, there are globally operating applications and websites that connect users all over the world, such as Etsy, Amazon and BeForward.

Online applications, which enable users to get their desired products through the exchange, are also quite popular in transactions between private users (P2P), for example, Rukkola, UsedUp and SwapTree. Crowdfunding platforms are open for private individuals too, who can opt to support a company, e.g., in the field of agriculture, in exchange for interest.

### 5.2. *The Segmentation of Product-Service Systems Market*

In the CoE product-service provider systems, we identified several activities: transportation, accommodation, professional and personal services, online content, assets, finance and funds. Users can enjoy all the benefits of a product or a service without buying it, thanks to the services provided in the product-service system. In exchange for a set price, they get temporary access only, which is a smaller expense from the consumer's viewpoint. Simultaneously, it is the utilization of a free capacity from the service provider's point of view.

### 5.2.1. CC in the Product-Service System (B2B, B2P)

In the case of the CC product–service system, the service is provided by a business enterprise. The use of platforms lowers the costs of transactions, thus rendering the prices more competitive than those operating on a value chain's traditional principles.

The CC mobility industry refers to Zipcar, Lime, Bird, Jump, MolBubi, MolLimo, etc., which offer their rentable vehicle fleet through simple telephone applications. Several carmakers have recognized that users are not always willing or able to buy the more expensive new cars, especially when they want to use them for a short time only and do not want to cover the costs of maintenance, parking and insurance. Therefore, carmakers have started to develop—alone or in the form of joint ventures—applications that facilitate complex sharing with other companies, e.g., Drive Now, the application of BMW and Mini, Car2Go, the application of Daimler, or the car-sharing platform of Škoda, called Uniqway [55]. These carmakers have started using a new level of mobility, connectivity and digitalization.

The CC tourism and hotel industry service providers are the companies that usually operate hotels, too. Relevant applications include AccorHotels's onefinestay.com, which was purchased in 2016 to respond to the digital challenge posed by the booking application Airbnb. In 2017 Hyatt did the same by investing in an Oasis application, where renters can choose from unique offers (Unbound Collection) of exclusive flats and apartments, enjoying all the advantages of a hotel stay.

CC professional and personal services are represented by online mediator platforms. When the service provider is a company, the Internet platform where the company appears usually refers to a sphere of activities, for example, cleaning (Rendi, Budapest, Hungary)

or transport (AmazonFlex, Seattle, WA, USA). Companies that operate co-working offices belong to this category, such as Loffice Coworking (Budapest, Hungary) or Xpotential (Debrecen, Hungary), where startups can use high level infrastructural and administrative backgrounds they usually cannot afford. Companies do not need these services; they are considered an alternative to the home office.

Regarding CC entertainment, multimedia and telecommunication (online content), we would only like to mention the most extensive examples, such as Netflix, Prime Video, Spotify and Deezer. Each is a streaming service; consumers get access to their online stores by paying a monthly fee. These online content services offer entertainment and popular online courses; take, for example, LinkedIn's Lyndra or Khan Academy's coursebook library [2,37,51,56]. Also, increasingly tertiary education institutions recognize that they can profit from their professors' and researchers' knowledge with the help of CoE.

In the financial sector (finance) of the product-service system, disruptive technologies have led to a significant performance improvement while reducing costs. Nowadays, financial services are not offered exclusively by bank institutions since new actors and models have appeared in financial transactions, be it lending or transfer services, due to the decentralized state of information and communication technology. Belleflamme et al. [77] distinguished two significant types of crowdfunding platforms (CFP), such as investment-based and reward- and donation-based platforms (Figure 3).

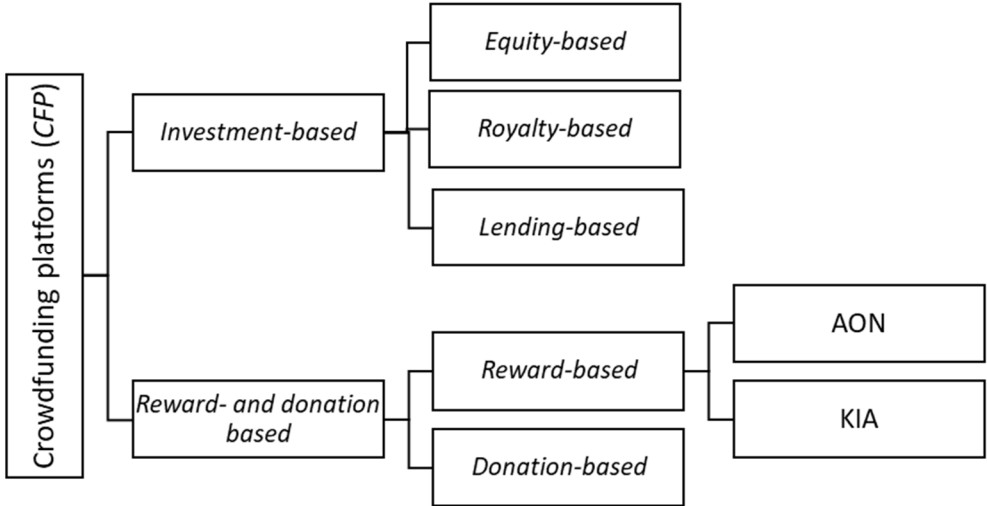

**Figure 3.** The classification of crowdfunding platforms; Source: Authors' elaboration based on Belleflamme et al. [77].

The investment-based CFP has the following subgroups:

- Equity-based finance, where the entrepreneur offers ownership in exchange for financing. We have already introduced this financing model in connection with redistribution markets.
- Royalty–based finance, where the project promoter offers investors a share of future royalties generated from the fund; e.g., Quirky.
- Lending–based finance, where the community members provide a loan to a person or a company, like CircleUp, MagNetBank, Prosper, Zopa, Grupeer, FundingCircle, Lending Club.
- In exchange for a reward- and donation-based finance, investors do not expect financial gain because they get services or products, and/or they subsidize the creation of these. The reward- and donation-based CFP has the following subgroups:
- All-Or-Nothing (AON) finance model, where the entrepreneur predetermines the desired amount of money they will receive once it is collected, so the project implementation starts when the limit sum is reached; e.g., Kickstarter, Crowdfunder, Crowdsupply, WeAreHere, Multifintare.ro, Potsieu.ro, Startarium.

- Keep-It-All (KIA) finance model, where the sum can be used to implement the project, even if it has not reached the predetermined amount, e.g., IndieGogo, GoFundMe, Sponsume.
- Donation-based finance, where social, community-related, creative or personal projects are subsidized, in the form of occasional or regular donations, for which donors expect nothing in return [78], e.g., Causes, AdjukOssze.hu, Patreon.

The crowdfunding models presented above can belong to both the CC and the SE product-service systems. They cannot be connected categorically to companies or private individuals, so we introduced them only here.

In the CC product-service systems, consumer goods refer to an activity of business enterprises in which companies lend their idle capacity of heavy equipment to make better use of them, e.g., in agriculture (HelloTracto.com) and the construction industry (Edilmag). The renters could order the requisite equipment through an application in which they can also carry out the payment (with FinTech systems) and the insurance (via insurtech methods) and settle claims.

### 5.2.2. SE in the Product-Service Systems (P2B, P2P)

In the product-service system, the SE is the category most authors whose studies we analyzed identify as the SE, based on their definition criteria [7,8,36,37,54]. In the SE product–service system, the service providers are private individuals (peers) who desire to share their goods or services with consumers in a limited number in exchange for a fee. Even in this case, the intermediary is an Internet platform through which customers and vendors can find each other.

In the SE mobility industry, better known as ride-sharing services, a private individual owning a car offers the vehicle's free seats for a set price. Relevant applications are, for example, Oszkár and BlaBlaCar. The car owner looks for passengers for a prearranged (date and destination) journey that the owner announces with the application's help, mainly intending to reduce travelling costs. This category includes the Uber application, which many people call a "taxi application" and offers passenger transport services for money and under strict rules evaluated by the riders. Similar applications are GetTransfer, Wunder Mobility and Grab.

The applications that belong to the SE tourism and hotel industry (e.g., Airbnb, HomeAway, Wimdu and 9flats) enable private individuals to rent out their temporarily unused properties, usually for a short period. The applications that deal with social parking (Barking, ParkOnMyRide and JustPark) also belong to this group. Social parking is a real-time service that allows registered users to find the nearest free or rentable parking places.

The appearance of the SE professional and personal services in applications breaks with traditional enterprises' operational norms, because in this case, private individuals, who mediate their services online, are integrated into the communities. Other names for this group are the on-demand economy and the gig economy. An on-demand service, which is announced in applications or on Internet platforms, means that consumers pay for the service based on the duration they use it for and only when they use it. Simultaneously, the word gig highlights that the service is occasional and used in the short run.

Such applications facilitate mainly short–term or occasional employment, from dog-walking to complex designing, e.g., Jószaki (hiring general handymen), TaskRabbit (hiring freelancers), Wag! (dog-walking), FoodPanda (delivery of food and many other things). Making the payment and giving feedback, and evaluating the service is possible through the platform itself.

The SE entertainment, multimedia and telecommunication group includes applications and websites where registered users share information; for example, the social mapping/traffic services of Waze provide real-time traffic data to travelers, edited by the Waze users themselves. Facebook, TikTok and YouTube offer numerous videos for education, entertainment and other purposes, also edited by the users.

In the case of the SE product-service systems, consumer goods refer to the distribution of goods that their owners do not need for a while and want to put at someone's disposal for money or for free. The classic example is the electric drill, used for a maximum of 10–15 min in a household and spends the rest of its life on a shelf [9]. The sharing of means reduces overconsumption, increases their utilization, and can generate extra income too. For this, one only needs to download an application and immediately start sharing their idle goods.

## 6. Limitations

This study, especially the classification modelling format process, is primarily based on the literature that categorized the CoE's different types. We mentioned trust as an essential feature of CoE, but we did not discuss it in more detail. According to Coase [79], trust has the effect of reducing transaction costs, and the lack of institutional background can be redeemed with confidence [80], which is also based on trust. The reduced transaction cost and the lack of regulation are characterizing CoE. We mention the pipeline-type business model in the study, but we do not discuss it in more detail. We would like to underline only the structural differences between the pipeline businesses and the platform-based businesses. The elaborated classification model does not reflect the for-profit and non-profit dimension of the CoE.

Furthermore, the CoE exists without using a platform, but these were not considered part of the demand-side, platform-based business model. However, we know that offline organizations and services represent another significant dimension of CoE practice. In explaining the demand-side, platform-based CoE business model's core activities, we illustrate their characteristics with case examples, and we did not aim to present all of them.

This study aims to define the concept and integrate it into society as its authors claim that these tasks are only partly achieved.

## 7. Conclusions

The demand-side platform-based CoE has been a part of everyday life for a decade in various forms. This platform-based business has been given various names based on its perceived attributes, which are not universal and overlaps often occur. After reviewing the literature of the last ten years, this article considered the various forms of the CoE, based on a new universal principle of systematization, and defined the demand-side and platform-based CoE and created a clear classification taxonomy.

The CoE is a demand-side, platform-based economic activity generated by consumers, built on trust and operated by information technology organizations in digital markets. The consumers' needs are met on an online Internet platform by providing immediate access to new or underutilized goods, services or money, with or without the transfer of ownership. Information is provided by multiple third parties on digital marketplace product or service, whereas the marketplace operator enables direct interaction between the participant groups.

CoE has two fundamental characteristics: (1) demand-side and (2) platform-based. This business model involves unlocking the commercial value by lending and reusing idle goods through selling and online markets for new products. We narrow down CoE into two categories: redistribution market and product-service system market. The difference between these two categories is the transfer of ownership or the lack of ownership transfer. Both the redistribution market and product-service systems can be further divided into two subgroups based on who is responsible for providing the service—a legal business entity or an individual (peer). At that point, we introduced the two subcategories of CoE, the SE and the CC. The difference between the SE and CC is the person or quality of the service provider. In the SE, the service provider is a private individual and transactions are executed in P2P and P2B configurations. In contrast, in CC, the service provider is a business enterprise, i.e., a legal person, typically profit-oriented, and transactions are executed in B2B and B2P configurations. Nonetheless, the implementation practice is

the same in both cases: a group is formed to respond to a business opportunity using digital platforms.

Based on our clear classification taxonomy, free from overlaps, we have elaborated a theoretical framework of the demand-side, platform-based CoE and labelled the subgroups. This labelling is essential to specify the targets for policies, strategies and studies. As a practical implication, using our classification framework, legislators could ensure that the SE and CC, as subsets of CoE, remain fair and create opportunities for all participants to thrive by rethinking the taxation and subsidizing policies and preparing national and international consultations and conventions. Most P2P and P2B transactions require a different approach than B2B and B2P transactions. The former involves sharing or selling already used and taxed goods or services, unlike B2B and B2P transactions. P2P and P2B transactions need more support for several reasons, while B2B and B2P transactions are normally taxable. In the case of P2P, P2B transactions, the seller or lessor is an individual, and the product which is the subject of exchange has been taxed several times formerly; e.g., personal income tax, sales tax, local government tax, etc. Therefore, the supplier (peer) paid for society's expectations for owning all of the goods. Once peer releases the taxed product back into the business cycle again, it will no longer have, e.g., neither its negative footprint on the natural environment nor its multi-stage transaction cost, e.g., supplier income tax, producer income tax, transportation income tax, sales income tax, etc. If only the latter are taken into account, it is already understandable that society should not tax but support SE transactions. Due to the above, SE economic activities contribute to the triple expectation of sustainability, namely environmental, economic and social. Of course, several further research is needed to explore all of these rigorously.

Our research revisited and analyzed the previous literature associated with describing the phenomena of the CoE. A significant limitation of previous work in this area is the lack of a clear consensus on categorizing and referring to various aspects of the CoE as much of the previous research has been ambiguous, fragmented and repetitive. This often has led to confusion and likely has even suppressed more rigorous study of the topic and has created gaps in the literature on both understanding and explaining the totality of the platform-based economy. We address these issues through this paper's principle contribution by presenting a new taxonomy for classifying aspects of the CoE that should aid researchers and managers alike. We hope that through using this newly created clearer framework, researchers will be better able to undertake more complex and diverse research studies about the CoE.

**Author Contributions:** Conceptualization, T.Z.K. and F.D.; Methodology, A.N. (András Nábrádi); Project administration, A.N. (Adrián Nagy) and I.S.; Supervision, A.N. (András Nábrádi); Validation, A.N. (András Nábrádi); Writing—original draft, T.Z.K. Writing—review & editing, T.Z.K., F.D. and A.N. (András Nábrádi). All authors have read and agreed to the published version of the manuscript.

**Funding:** The publication is supported by EFOP-3.6.3-VEKOP-16-2017-00007—Young researchers for talent—supporting careers in research activities in higher education program.

**Institutional Review Board Statement:** Not applicable.

**Informed Consent Statement:** Not applicable.

**Data Availability Statement:** Data sharing not applicable.

**Conflicts of Interest:** The author declare no conflict of interest.

## Appendix A

| No. | Related Articles and Studies | Platform Business Model | Core Activities | Market Structure | Mode of Exchange | New Business Model |
|-----|------------------------------|-------------------------|-----------------|------------------|------------------|--------------------|
| 1–2 | Kenney—Zysmann [1,81] | ✓ | - | - | - | - |
| 3 | Schor [2] | ✓ | ✓ | ✓ | ✓ | - |
| 4 | Sijabat [3] | ✓ | ✓ | - | - | - |
| 5 | Tabcum [4] | ✓ | - | ✓ | - | - |
| 6 | Owyang [5] | ✓ | ✓ | ✓ | - | - |
| 7 | Wirtz et al. [6] | ✓ | ✓ | ✓ | ✓ | ✓ |
| 8 | Gansky [12] | ✓ | ✓ | - | - | - |
| 9 | Hamari et al. [10] | ✓ | - | - | ✓ | - |
| 10 | Schor—Fitzmaurice [11] | ✓ | ✓ | - | - | - |
| 11 | Valant (EC) [8] | ✓ | ✓ | ✓ | ✓ | - |
| 12 | Ranjbari et al. [7] | ✓ | - | ✓ | ✓ | - |
| 13 | Kumar et al. [17] | ✓ | - | - | ✓ | - |
| 14 | PwC [24] | ✓ | ✓ | ✓ | - | - |
| 15 | Martin [16] | ✓ | - | - | ✓ | - |
| 16 | Sundararajan [28] | ✓ | ✓ | ✓ | ✓ | - |
| 17 | Friedman [29] | ✓ | ✓ | - | - | - |
| 18 | Gyulavári [30] | ✓ | ✓ | - | - | - |
| 19 | Eckhardt—Bardhi [31] | ✓ | - | ✓ | - | - |
| 20 | Cockayne [32] | ✓ | ✓ | - | - | - |
| 21 | Selloni [33] | ✓ | - | - | ✓ | - |
| 22 | Acquier et al. [13] | ✓ | ✓ | ✓ | ✓ | ✓ |
| 23 | Dudás—Boros [36] | ✓ | ✓ | ✓ | - | - |
| 24 | Botsman—Rogers [37] | ✓ | ✓ | ✓ | ✓ | ✓ |
| 25 | Van Alstyne et al. [44] | ✓ | ✓ | - | - | - |
| 26 | Hagiu—Wright [45] | ✓ | - | - | - | - |
| 27 | Sutherland—Jarrahi [46] | ✓ | - | - | ✓ | - |
| 28 | Bakó—Horváth [47] | ✓ | - | - | - | - |
| 29 | Johnson [50] | ✓ | - | - | - | - |
| 30 | Heinrichs [26] | ✓ | ✓ | - | ✓ | - |
| 31 | Stephany [52] | ✓ | ✓ | - | - | - |
| 32 | Codagnone et al. [53] | ✓ | ✓ | ✓ | ✓ | - |
| 33 | Szegedi [54] | ✓ | - | ✓ | ✓ | - |
| 34 | Haase—Pick [55] | ✓ | ✓ | - | ✓ | - |
| 35 | McDonald [56] | - | ✓ | - | ✓ | - |
| 36 | Rauch—Schleicher [57] | ✓ | ✓ | - | ✓ | - |
| 37 | Richardson [58] | ✓ | ✓ | - | ✓ | - |
| 38 | Winterhalter et al. [59] | - | - | - | ✓ | - |
| 39 | Edbring et al. [60] | ✓ | - | - | ✓ | - |
| 40 | Belk [61] | ✓ | - | - | ✓ | - |
| 41 | Bardhi—Eckhardt [63] | ✓ | - | - | ✓ | - |
| 42 | Matzler et al. [64] | ✓ | - | - | ✓ | - |
| 43 | Frenken— Schor [65] | ✓ | ✓ | ✓ | ✓ | ✓ |
| 44 | Gerwe—Silva [66] | ✓ | ✓ | ✓ | ✓ | - |
| 45 | Frenken [67] | ✓ | ✓ | ✓ | ✓ | ✓ |
| 46 | Plewnia—Guenther [68] | ✓ | ✓ | ✓ | - | - |
| 47 | Laurentini et al. [69] | ✓ | ✓ | ✓ | - | - |
| 48 | Choi et al. [70] | ✓ | - | - | ✓ | - |
| 49 | Olson—Kemp [71] | ✓ | ✓ | ✓ | - | - |
| 50 | Belleflamme et al. [77] | ✓ | ✓ | ✓ | - | - |

| No. | Related Articles and Studies | Platform Business Model | Core Activities | Market Structure | Mode of Exchange | New Business Model |
|---|---|---|---|---|---|---|
| 51 | Kuti—Madarász [78] | √ | √ | √ | - | - |
| 52 | Pelzer—Burgard [82] | √ | √ | - | - | - |
| 53 | Cohen—Munoz [83] | √ | √ | √ | - | - |
| 54 | Rifkin [84] | √ | √ | - | - | - |
| 55 | Munoz—Cohen [85] | √ | - | √ | - | - |
| 56 | Möhlmann [86] | √ | - | - | √ | - |
| 57 | Lamberton—Rose [87] | √ | √ | √ | - | - |
| 58 | Cohen—Kietzmann [88] | √ | √ | √ | - | - |
| 59 | Benoit et al. [89] | √ | √ | √ | - | - |

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
