# Peer review of "An Analysis of the Demand-Side, Platform-Based Collaborative Economy: Creation of a Clear Classification Taxonomy"

_sustainability, doi:10.3390/su13052817_

Round 1

Reviewer 1 Report

This is a paper which contributes to a topic which has actually received low attention, but has potential for future research. My comments are minor.

1. Thank you for the opportunity to read this interesting paper. I think the paper presents interesting models but unfortunately, at theoretically the paper is still very underdeveloped.

Changes which must be made before publication:

1. The literature used in this paper is quite varied and up-to-date – However, insufficient. Please refer to CE.

The authors should supplement it with:

Fonseca, et al., (2018). Assessment of circular economy within Portuguese organizations. Sustainability, 10(7), 2521.

Geissdoerfer, M., Savaget, P., Bocken, N. M., & Hultink, E. J. (2017). The Circular Economy–A new sustainability paradigm?. Journal of cleaner production, 143, 757-768.

etc.

2. Please describe in more detail how you carried out the research process. I suggest adding a drawing in which you will show the stages of the research process in turn.

3. The implications for research, theory, practice and society are not clear though I can see that these aspects can be elaborated further. Research limitations must be explained.

Good Luck !

Author Response

The introduction has been supplemented (lines 45-49) to contextualize the theoretical background better. In line 78, Table 1 has been expanded to include Circular Economy as an alternative to the phrase sharing economy. We expanded the literature background to improve the theoretical background (from 50 to 86 references). In-text references follow this expansion, of course.

The requested references were included in the study (ref. 15 and 16)

In the methodological section, following Paul et al. Method (TCM Method)  was incorporated in (table 2), where the research process steps were indicated (133 - 150 lines): research objectives, initial inclusion criteria, setting the exclusion criteria, applying the exclusion criteria, content analysis and discussion and conclusion. In support of these, Appendix 1 has been prepared, which lists the literature that meets the exclusion criteria.

Research limitation was incorporated and explained in the study (lines 684–696).

Reviewer 2 Report

I would suggest the publication of this article.

Author Response

Thank you for your kind review!

Reviewer 3 Report

The article suffers from critical methodological problems - the authors state that "This study examines the articles published during the previous ten years, which have aimed not only to define but also to classify the collaborative economy according to various criteria".

However, there is no info how examined articles have been selected, moreover the literature list does not show comprehensive coverage of the topic. It is also unclear, how - on which base the so called "logical scheme" has been constructed:
1. The separation of the pipeline business model from platform-based, demand side business models.
2. The definition of the platform–based, demand-side business model.
3. The categorization of the platform–based, demand-side business model.

Author Response

We expanded the literature background to improve the theoretical background (from 50 to 86 references). In-text citations follow this expansion, of course.

We updated the Methodology section. Following Paul et al. method (TCM Method), were incorporated Table 2, where the research process scheme was indicated (133 - 150 lines): research objectives, initial inclusion criteria, setting the exclusion criteria, applying the exclusion criteria, content analysis and discussion and conclusion.  In support of these, Appendix 1 has been prepared, containing the literature that meets the exclusion criteria. With such additions, the logical scheme of the research was also presented.

Round 2

Reviewer 3 Report

OK

Author Response

Thank you.